# EVALUATING SEMANTIC REPRESENTATIONS OF SOURCE CODE

## ABSTRACT

Learned representations of source code enable various software developer tools, e.g., to detect bugs or to predict program properties. At the core of code representations often are word embeddings of identifier names in source code, because identifiers account for the majority of source code vocabulary and convey important semantic information. Unfortunately, there currently is no generally accepted way of evaluating the quality of word embeddings of identifiers, and current evaluations are biased toward specific downstream tasks. This paper presents IdBench, the first benchmark for evaluating to what extent word embeddings of identifiers represent semantic relatedness and similarity. The benchmark is based on thousands of ratings gathered by surveying 500 software developers. We use IdBench to evaluate state-of-the-art embedding techniques proposed for natural language, an embedding technique specifically designed for source code, and lexical string distance functions, as these are often used in current developer tools. Our results show that the effectiveness of embeddings varies significantly across different embedding techniques and that the best available embeddings successfully represent semantic relatedness. On the downside, no existing embedding provides a satisfactory representation of semantic similarities, e.g., because embeddings consider identifiers with opposing meanings as similar, which may lead to fatal mistakes in downstream developer tools. IdBench provides a gold standard to guide the development of novel embeddings that address the current limitations.

## 1 INTRODUCTION

Reasoning about source code based on learned representations has various applications, such as predicting method names (Allamanis et al., 2015), detecting bugs (Pradel & Sen, 2018) and vulnerabilities (Harer et al., 2018), predicting types (Malik et al., 2019), detecting similar code (White et al., 2016; Xu et al., 2017), inferring specifications (DeFreez et al., 2018), code de-obfuscation (Raychev et al., 2015; Alon et al., 2018a), and program repair (Devlin et al., 2017). Many of these techniques are based on embeddings of source code, which map a given piece of code into a continuous vector representation that encodes some aspect of the semantics of the code.

A core component of most code embeddings are semantic representations of identifier names, i.e., the names of variables, functions, classes, fields, etc. in source code. Similar to words in natural languages, identifiers are the basic building block of source code. Identifiers not only account for the majority of the vocabulary of source code, but they also convey important information about the (intended) meaning of code. To reason about identifiers and their meaning, code analysis techniques build on learned embeddings of identifiers, either by adapting embeddings that were originally proposed for natural languages (Mikolov et al., 2013a;b) or with embeddings specifically designed for source code (Alon et al., 2018a).

Given the importance of identifier embeddings, a crucial challenge is measuring how effective an embedding represents the semantic relationships between identifiers. For word embeddings in natural language, the community has addressed this question through a series of gold standards (Finkelstein et al., 2002; Bruni et al., 2014a; Rubenstein & Goodenough, 1965; Miller & Charles, 1991; Hill et al., 2015; Gerz et al., 2016). These gold standards define how similar two words are based on ratings by human judges, enabling an evaluation that measures how well an embedding reflects the human ratings.

Unfortunately, simply reusing existing gold standards to identifiers in source code would be misleading. One reason is that the vocabularies of natural languages and source code overlap only partially, because source code contains various terms and abbreviations not found in natural language texts. Moreover, source code has a constantly growing vocabulary, as developers tend to invent new identifiers, e.g., for newly emerging application domains Babii et al. (2019). Finally, even words present in both natural languages and source code may differ in their meaning due to computer science-specific meanings of some words, e.g., "float" or "string".

This paper addresses the problem of measuring and comparing the effectiveness of embeddings of identifiers. We present IdBench, a benchmark for evaluating techniques that represent semantic similarities of identifiers. The basis of the benchmark is a dataset of developer opinions about the similarity of pairs of identifiers. We gather this dataset through two surveys that show real-world identifiers and code snippets to hundreds of developers, asking them to rate their similarity. Taking the developer opinions as a gold standard, IdBench allows for evaluating embeddings in a systematic way by measuring to what extent an embedding agrees with ratings given by developers. Moreover, inspecting pairs of identifiers for which an embedding strongly agrees or disagrees with the benchmark helps understand the strengths and weaknesses of current embeddings.

Overall, we gather thousands of ratings from 500 developers. Cleaning and compiling this raw dataset into a benchmark yields several hundreds of pairs of identifiers with gold standard similarities, including identifiers from a wide range of application domains. We apply our approach to a corpus of JavaScript code, because several recent pieces of work on identifier names and code embeddings focus on this language (Pradel & Sen, 2018; Alon et al., 2018b;a; Malik et al., 2019). Applying our methodology to another language is straightforward.

Based on the newly created benchmark, we evaluate and compare state-of-the-art embeddings of identifiers. We find that different embedding techniques differ heavily in terms of their ability to accurately represent identifier relatedness and similarity. The best available technique, the CBOW variant of FastText, accurately represents relatedness, but none of the available techniques accurately represents identifier similarities. One reason is that some embeddings are confused about identifiers with opposite meaning, e.g., `rows` and `cols`, and about identifiers that belong to the same application domain but are not similar. Another reason is that some embeddings miss synonyms, e.g., `file` and `record`. We also find that simple string distance functions, which measure the similarity of identifiers without any learning, are surprisingly effective, and even outperform some learned embeddings for the similarity task.

In summary, this paper makes the following contributions. (1) *Methodology*: To the best of our knowledge, we are the first to systematically evaluate embeddings of identifiers. Our methodology is based on surveying developers and summarizing their opinions into gold standard similarities of pairs of identifiers. (2) *Reusable benchmark*: We make available a benchmark of hundreds of pairs of identifiers, providing a way to systematically evaluate existing and future embeddings.[1] (3) *Comparison of state-of-the-art embeddings*: We evaluate seven existing embeddings and string similarity functions, and discuss their strengths and weaknesses.

## 2 BENCHMARK DESIGN

We design IdBench so that it distinguishes two kinds of semantic relationships between identifiers (Miller & Charles, 1991; Hill et al., 2015; Faruqui et al., 2016). On the one hand, *relatedness* refers to the degree of association between two identifiers and covers various possible relations between them. For example, `top` and `bottom` are related because they are opposites, `click` and `dblclick` are related because they belong to the same general concept, and `getBorderWidth` and `getPadding` are related because they belong to the same application domain. On the other hand, *similarity* refers to the degree to which two identifiers have the same meaning, in the sense that one could substitute the other (Miller & Charles, 1991) without changing the overall meaning. For example, `length` and `size`, as well as `username` and `userid`, are similar to each other. Similarity is a stronger semantic relationship than relatedness, because the former implies the latter, but not vice-versa. For example, the identifiers `start` and `end` are related, as they are opposites,

---

[1]`https://my.pcloud.com/publink/show?code=kZR2LzkZAViNngFuJ3ykFqNzbQF1TJRedJs7`

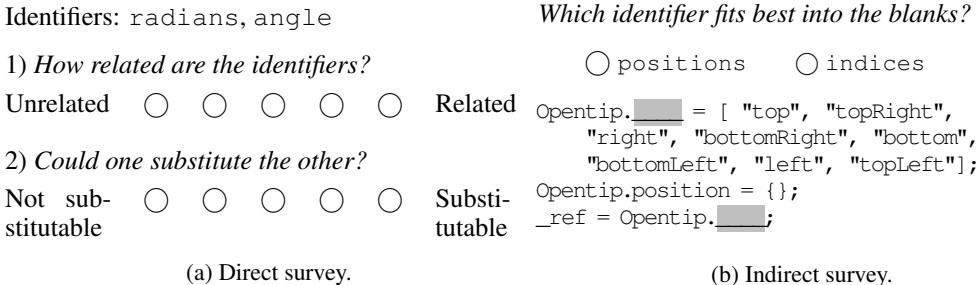

Figure 1: Examples of the developer surveys.

but they are not similar, because replacing one with the other would change the meaning of the program.

IdBench includes three benchmark tasks: A *relatedness task* and two task to measure how well an embedding reflects the similarity of identifiers: a *similarity task* and a *contextual similarity task*. The following describes how we gather developer opinions that provide data for these tasks.

## 2.1 DEVELOPER SURVEYS

**Direct Survey of Developer Opinions**   This survey shows two identifiers to a developer and then directly asks how related and how similar the identifiers are. Figure 1a shows an example question from the survey. The developer is shown pairs of identifiers and is then asked to rate on a five-point Likert scale how related and how similar these identifiers are to each other. In total, each developer is shown 18 pairs of identifiers, which we randomly sample from a larger pool of pairs. Before showing the questions, we provide a brief description of what the developers are supposed to do, including an explanation of the terms "related" and "substitutable". The ratings gathered in the direct survey are the basis for the relatedness task and the similarity task of IdBench.

**Indirect Survey of Developer Opinions**   This survey asks developers to pick an identifier that best fits a given code context, which indirectly asks about the similarity of identifiers The motivation is that identifier names alone may not provide enough information to fully judge how similar they are (Miller & Charles, 1991). For example, without any context, identifiers idx and hl may cause confusion for developers who are trying to judge their similarity. The survey addresses this challenge by showing the code context in which an identifier occurs, and by asking the developers to decide which of two given identifiers best fits this context. If, for a specific pair of identifiers, the developers choose both identifiers equally often, then the identifiers are likely to be similar to each other, since one can substitute the other. Figure 1b shows a question asked during the indirect survey. As shown in the example, for code contexts where the identifier occurs multiple times, we show multiple blanks that all refer to the same identifier. In total, we show 15 such questions to each participant of the survey. The ratings gathered in the indirect survey are the basis for the contextual similarity tasks of IdBench.

**Selection of Identifiers and Code Examples**   We select identifiers and code contexts from a corpus of 50,000 JavaScript files (Raychev et al., 2016). We select 300 pairs, made out of 488 identifiers, through a combination of automated and manual selection, aimed at a diverse set that covers different degrees of similarity and relatedness. The selection is guided by similarities between identifiers as judged by word embeddings Bruni et al. (2014b). The first step is to extract from the code corpus all identifier names that appear more than 100 times, which results in about 17,000 identifiers, including method names, variable names, property names, other types of identifiers. Next, we compute word embeddings of the extracted identifiers, compute the cosine distance of all pairs of identifiers, and then sort the pairs according to their cosine distances. We use two word embeddings Mikolov et al. (2013a); Alon et al. (2018a), giving two sorted lists of pairs, and sample 150 pairs from each list as follows: (i) Pick 75 of the first 1,000 pairs, i.e., the most similar pairs according to the respective embedding, where 38 pairs are randomly sampled and 37 pairs are manually selected. We manually select some pairs because otherwise, we had observed a lack of pairs of synonymous identifiers. (ii)

Randomly sample 50 from pairs 1,001 to 10,000, i.e., pairs that still have a relatively high similarity. (iii) Randomly sample 25 pairs from the remaining pairs, i.e., pairs that are mostly dissimilar to each other.

To gather the code contexts for the indirect survey, we search the code corpus for occurrences of the selected identifiers. As the size of the context, we choose five lines, which provides sufficient context to choose the best identifier without overwhelming developers with large amounts of code. For each identifier, we randomly select five different contexts. When showing a specific pair of identifiers to a developer, we randomly select one of the gathered contexts for one of the two identifiers.

**Participants** We pay developers via Amazon Mechanical Turk to perform the surveys. Participants take around 15 minutes to complete both surveys. In total, 500 developers participate in the survey, which yields at least 10 ratings for each pair of identifiers.

## 2.2 DATA CLEANING

To eliminate noise in the gathered data, e.g., due to lack of expertise or involvement by the participants, we clean the data by removing some participants and identifier pairs.

**Removing Outliers** As a first filter, we remove outlier participants based on the Inter-Rater Agreement (IRA), which measures the degree of agreement between participants. We use Krippendorf's alpha coefficient, because it handles unequal sample sizes, which fits our data, since not all participants rate the same pairs, and because not all pairs have the same number of ratings. The coefficient ranges between zero and one, where zero represents complete disagreement and one represents perfect agreement. For each participant, we calculate the difference between her rating and the average of all the other ratings for each pair. Then, we average these differences for each rater, and discard participants with a difference above a threshold. We perform this computation both for the relatedness and similarity ratings from the direct survey, and then remove outliers based on the average difference across both ratings.

**Removing Downers** As a second filter, we eliminate participants that decrease the overall IRA, called *downers* (Hill et al., 2015), because they bring the agreement level between all participants down. For each participant $p$, we compute $IRA_{sim}$ and $IRA_{rel}$ after removing the ratings of $p$ from the data. If $IRA_{sim}$ or $IRA_{rel}$ increases by at least 10%, then we discard that participant's ratings.

All IRA-based filtering of participants is based on ratings from the direct survey only. For the indirect survey, computing the IRA would be inadequate because the code contexts shown along with a pair of identifiers are randomly sampled, i.e., a pair is rarely shown with the same context. Instead, we exploit the fact that the two surveys are shown in sequence to each participant. Specifically, we discard participants based on the first two filters for both the direct and indirect survey, assuming that participants discarded in the direct survey are not worth keeping in the indirect survey.

**Removing Pairs with Confusing Contexts** As a third filter, we eliminate some pairs of identifiers used in the indirect survey. Since our random selection of code contexts may include contexts that are not helpful in deciding on the most suitable identifier, the ratings for some pairs are rather arbitrary. To mitigate this problem, we remove a pair if the difference in similarity as rated in the direct and indirect surveys exceeds some threshold.

Table 2 shows the number of identifier pairs that remain in the benchmark after data cleaning. For each of the three tasks, we provide a small, medium, and large benchmark, which differ by the thresholds used during data cleaning. The smaller benchmarks use stricter thresholds and hence provide higher agreements between the participants, whereas the larger benchmarks offer more pairs.

## 2.3 GOLD STANDARD OF SIMILARITY SCORES

**Converting Ratings to Scores** To ease the evaluation of embeddings, we convert the ratings gathered for a specific pair during the developer surveys into a similarity score in the $[0, 1]$ range. For the direct survey, we scale the 5-point Likert-scale ratings into the $[0, 1]$ range and average all ratings for

Figure 2: Benchmark sizes and inter-rater agreement.

| Size | Task | | | | |
|---|---|---|---|---|---|
| | Relatedn. | | Simil. | | Context. simil. |
| | Pairs | IRA | Pairs | IRA | Pairs |
| Small | 167 | 0.67 | 167 | 0.62 | 115 |
| Medium | 247 | 0.61 | 247 | 0.57 | 145 |
| Large | 291 | 0.56 | 291 | 0.51 | 176 |

Figure 3: Pairs with gold standard similarities.

| Identifier 1 | Identifier 2 | Score | | |
|---|---|---|---|---|
| | | Rel. | Simil. | Cont. simil. |
| substr | substring | 0.94 | 1.00 | 0.89 |
| setMinutes | setSeconds | 0.91 | 0.22 | 0.06 |
| reset | clear | 0.90 | 0.89 | 0.94 |
| rows | columns | 0.88 | 0.08 | 0.22 |
| setInterval | clearInterval | 0.86 | 0.09 | 0.34 |
| count | total | 0.83 | 0.81 | 0.79 |
| item | entry | 0.78 | 0.77 | 0.92 |
| miny | ypos | 0.68 | 0.37 | 0.02 |
| events | rchecked | 0.16 | 0.14 | 0.18 |
| re | destruct | 0.06 | 0.02 | 0.02 |

a specific pair of identifiers. For the indirect survey, we use a signal detection theory-based approach for converting the collected ratings into numeric values (Miller & Charles, 1991). This conversion yields an unbounded distance measure $d$ for each pair, which we convert into a similarity score $s$ by normalizing and inverting the distance: $s = 1 - \frac{d - min_d}{max_d - min_d}$, where $min_d$ and $max_d$ are the minimum and maximum distances across all pairs.

**Examples** Table 3 shows representative examples of identifier pairs and their scores for the three benchmark tasks. The example illustrates that the scores match human intuition and that the gold standard clearly distinguishes relatedness from similarity. Some of the highly related and highly similar pairs, e.g., `substr` and `substring`, are lexically similar, while others are synonyms, e.g., `count` and `total`. While identifiers like `rows` and `columns` are strongly related, one cannot substitute the other, and they hence have low similarity. Similarly `miny`, `ypos` represent distinct properties of the variable `y`. Finally, some pairs are either weakly or not at all related, e.g., `re` and `destruct`.

## 3 EVALUATION OF WORD EMBEDDINGS FOR IDENTIFIERS

**Embeddings and String Distance Functions** To assess how well existing embedding techniques represent the relatedness and similarity of identifiers, we evaluate five vector representations against IdBench. We evaluate (i) the continuous bag of words and the skip-gram variants of *Word2vec* (Mikolov et al., 2013a;b) ("w2v-cbow" and "w2v-sg"), because recent identifier-based code analysis tools, e.g., DeepBugs (Pradel & Sen, 2018) and NL2Type (Malik et al., 2019) use it, (ii) *FastText* (Bojanowski et al., 2017), a sub-word extension of Word2vec that represents words as character n-grams ("FT-cbow" and "FT-sg"), and (iii) an embedding technique specifically designed for code, which learns from paths through a structural, tree-based representation of code (Alon et al., 2018a) ("path-based"). We train all embeddings on the same code corpus of 50,000 JavaScript files Raychev et al. (2016). For each embedding, we experiment with various hyper-parameters (e.g., dimension, number of context words) and report results only for the best performing models.

In addition to neural embeddings of identifiers, we also evaluate two string distance functions: *Levenshtein's* edit distance and *Needleman-Wunsch* distance (Needleman & Wunsch, 1970). These functions use lexical similarity as a proxy for the semantic relatedness of identifiers. We consider these functions because they are used in identifier-based code analysis tools, including a bug detection tool deployed at Google (Rice et al., 2017).

**Measuring Agreement with the Benchmark** We measure the magnitude of agreement of an embedding with IdBench by computing Spearman's rank correlation $\rho$ between the cosine similarities of pairs of identifier vectors and the gold standard of similarity scores. For string similarity functions, we compute the similarity score $s = 1 - d_{norm}$ for each pair based on a normalization $d_{norm}$ of the distance returned by the function.

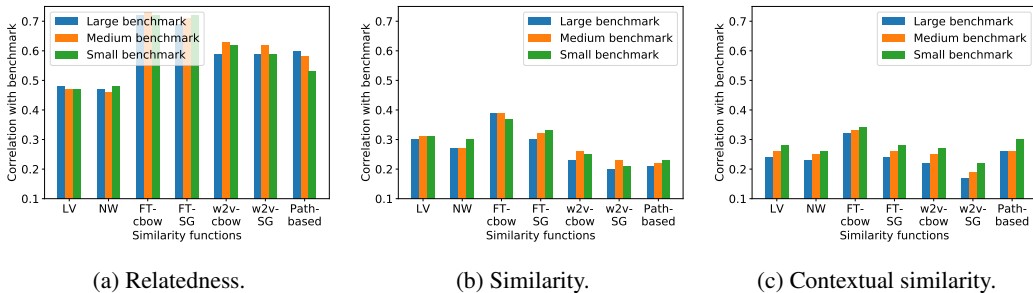

| (a) Relatedness. | (b) Similarity. | (c) Contextual similarity. |

Figure 4: Correlations of embeddings and string distance functions with the benchmark.

## 3.1 RESULTS

Figure 4 shows the agreement of the evaluated techniques with the small, medium, and large variants of IdBench. All embeddings and string distance functions agree to some extent with the gold standard. Overall, FastText-cbow consistently outperforms all other techniques, both for relatedness and similarity. We discuss the results in more detail in the following.

**Relatedness**  As shown in Figure 4a, all techniques achieve relatively high levels of agreement, with correlations between 46% and 73%. The neurally learned embeddings clearly outperform the string distance-based similarity functions (53-73% vs. 46-48%), showing that the effort of learning a semantic representation is worthwhile. In particular, the learned embeddings match or even slightly exceed the IRA, which is sometimes considered an upper bound of how strongly an embedding may correlate with a similarity-based benchmark (Hill et al., 2015). Comparing different embedding techniques with each other, we find that both FastText variants achieve higher scores than all other embeddings. In contrast, despite using additional structural information of source code, path-based embeddings score only comparably to Word2vec.

**Similarity**  Figure 4b shows a much lower agreement with the gold standard for similarity than for relatedness. One explanation is that encoding semantic similarity is a harder task than encoding the less strict notion of relatedness. Similar to relatedness, FastText-cbow shows the strongest agreement, ranging between 37% and 39%. A perhaps surprising result is string distance functions outperforming some of the embeddings.

**Contextual similarity**  The results of the contextual similarity task (Figure 4c), confirm the findings from the similarity task. All studied techniques are less effective than for relatedness, and FastText-cbow achieves the highest agreement with IdBench. One difference between the results for similarity and contextual similarity is that we here observe higher scores by path-based embeddings. We attribute this result to the ability of the path-based model to represent code snippets in a vector space.

## 3.2 DISCUSSION

While the best available embeddings are highly effective at representing relatedness, none of the studied techniques reaches the same level of agreement for similarity. In fact, even the best results in Figures 4b and 4c (39%) clearly stay beyond the IRA of our benchmark (62%), showing a huge potential for improvement. For many applications of embeddings of identifiers, semantic similarity is crucial. For example, tools to suggest suitable variable or method names (Allamanis et al., 2015; Alon et al., 2018a) aim for the name that is most similar, not only most related, to the concept represented by the variable or method. Likewise, identifier name-based tools for finding programming errors (Pradel & Sen, 2018) or variable misuses (Allamanis et al., 2017) want to identify situations where the developer uses a wrong, but perhaps related, variable. The lack of embeddings that accurately represent the semantic similarities of identifiers motivates more work on embedding techniques suitable for this task.

Table 1: Top-5 most similar identifiers by the FastText-cbow and path-based models.

| Identifier | Embedding | Nearest neighbors | | | | |
|---|---|---|---|---|---|---|
| substr | FT-cbow | substring | substrs | subst | substring1 | substrCount |
| | Path-based | substring | getInstanceProp | getPadding | getMinutes | floor |
| item | FT-cbow | itemNr | itemJ | itemL | itemI | itemAt |
| | Path-based | entry | child | record | targ | nextElement |
| count | FT-cbow | countTbl | countInt | countRTO | countsAsNum | countOne |
| | Path-based | total | limit | minVal | exponent | rate |
| rows | FT-cbow | rowOrRows | rowXs | rows_l | rowsAr | rowIDs |
| | Path-based | cols | cells | columns | tiles | items |
| setInterval | FT-cbow | resetInterval | setTimeoutInterval | clearInterval | getInterval | retInterval |
| | Path-based | clearInterval | assume | alert | nextTick | ReactTextComponent |
| minText | FT-cbow | maxText | minLengthText | microsecText | maxLengthText | minuteText |
| | Path-based | maxText | displayMsg | blankText | disableText | emptyText |
| files | FT-cbow | filesObjs | filesGen | fileSets | extFiles | libFiles |
| | Path-based | records | tasks | names | tiles | todos |
| miny | FT-cbow | min_y | minBy | minx | minPt | min_z |
| | Path-based | minx | ymin | dataMax | dataMin | ymax |

To better understand why current embeddings sometimes fail to accurately represent similarities, Table 1 shows the most similar identifiers of selected identifiers according to the FastText-cbow and path-based embeddings. The examples illustrate two observations. First, FastText, due to its use of n-grams (Bojanowski et al., 2017), tends to cluster identifiers based on lexical commonalities. While many lexically similar identifiers are also semantically similar, e.g., `substr` and `substring`, this approach misses other synonyms, e.g., `item` and `entry`. Another downside is that lexical similarity may also establish wrong relationships. For example, `substring` and `substrCount` represent different concepts, but FastText finds them to be highly similar.

Second, in contrast to FastText, path-based embeddings tend to cluster words based on their structural and syntactical contexts. This approach helps the embeddings to identify synonyms despite their lexical differences, e.g., `count` and `total`, or `files` and `records`. The downside is that it also clusters various related but not similar identifiers, e.g., `minText` and `maxText`, or `substr` and `getPadding`. Some of these identifiers even have opposing meanings, e.g., `rows` and `cols`, which can mislead code analysis tools when reasoning about the semantics of code.

A somewhat surprising result is that simple string distance functions achieve a level of agreement with IdBench's similarity gold standards as high as some learned embeddings. The reason why string distance functions sometimes correctly identify semantic similarities is that some semantically similar identifiers are also be lexically similar. One downside of lexical approaches is that they miss synonymous identifiers, e.g., `count` and `total`.

## 4 RELATED WORK

**Benchmarks of Word Embeddings**   The wide use of word embeddings in NLP raises the question of how to compare and evaluate word embeddings. Several gold standards based on human judgments have been proposed, focusing on either relatedness (Finkelstein et al., 2002; Bruni et al., 2014a; Schnabel et al., 2015) or similarity (Rubenstein & Goodenough, 1965; Miller & Charles, 1991; Hill et al., 2015; Gerz et al., 2016) of words. While these existing gold standards for NL words have been extremely inspiring, they are insufficient to evaluate embeddings of identifiers. One reason is that the vocabulary of source code contains various words not found in standard NL vocabulary, e.g., abbreviations and domain-specific terms, leading to very large vocabularies (Babii et al., 2019). Moreover, even identifiers found also in NL may have a different meaning in source code, e.g., `float` or `string`. This work is the first to address the need for a gold standard for identifiers.

**Data gathering**   Asking human raters how related or similar two words are was first proposed by Rubenstein & Goodenough (1965) and then adopted by others (Miller & Charles, 1991; Finkelstein et al., 2002; Hill et al., 2015; Gerz et al., 2016). Our direct survey also follows this methodology. Miller & Charles (1991) propose to gather judgments about contextual similarity by asking participants to choose a word to fill in a blank, an idea we adopt in our indirect survey. To choose words and pairs of words, prior work relies on manual selection (Rubenstein & Goodenough, 1965), pre-existing free association databases (Hill et al., 2015; Gerz et al., 2016), e.g., USF (Nelson et al., 2004) or VerbNet (Kipper et al., 2004; 2008), or cosine similarities according to pre-existing models (Bruni et al., 2014a;b). We follow the latter approach, as it minimizes human bias while covering a wide range of degrees of relatedness and similarity.

**Inter-rater agreement**   Gold standards for NL words reach an IRA of 0.61 (Finkelstein et al., 2002) and 0.67 (Hill et al., 2015). Our "small" dataset reaches similar levels of agreement, showing that the rates in IdBench represent a genuine human intuition. As noted by Hill et al. (2015), the IRA also gives an upper bound of the expected correlation between the tested model and the gold standard. Our results show that current models still leave plenty of room for improvement, especially w.r.t. similarity.

**Embeddings of identifiers**   Embeddings of identifiers are at the core of several code analysis tools. A popular approach, e.g., for bug detection (Pradel & Sen, 2018), type prediction (Malik et al., 2019), or vulnerability detection (Harer et al., 2018), is applying Word2vec (Mikolov et al., 2013a;b) to token sequences, which corresponds to the Word2vec embedding evaluated in Section 3. White et al. (2016) train an RNN-based language model and extract its final hidden layer as an embedding of identifiers. Chen & Monperrus (2019) provide a more comprehensive survey of embeddings for source code. Beyond learned embeddings, string similarity functions are used in other name-based tools, e.g., for detecting bugs (Pradel & Gross, 2011; Rice et al., 2017) or for inferring specifications (Zhong et al., 2009). The quality of embeddings is crucial in these and other code analysis tools, and IdBench will help to improve the state of the art.

**Embeddings of programs**   Beyond embeddings of identifiers, there is a stream of work on embedding larger parts of a program. Allamanis et al. (2015) use a log-bilinear, neural language model (Bengio et al., 2003) to predict the names of methods. Other work embeds code based on graph neural networks (Allamanis et al., 2017) or sequence-based neural networks applied to paths through a graph representation of code (Ben-Nun et al., 2018; Alon et al., 2019b; Devlin et al., 2017; Henkel et al., 2018; DeFreez et al., 2018; Xu et al., 2017). Code2seq embeds code and then generates sequences of NL words (Alon et al., 2019a). Allamanis et al. (2018) gives a detailed survey of learned models of code. To evaluate embeddings of programs, Wang & Christodorescu (2019) propose the COSET benchmark that provide thousands of programs with semantic labels. Ours and their work complement each other, as COSET evaluates embeddings of entire programs, whereas IdBench evaluates embeddings of identifiers. Since identifiers are a basic building block of source code, a benchmark for improving embeddings of identifiers will eventually also benefit learning-based code analysis tools.

## 5   CONCLUSION

This paper presents the first benchmark for evaluating vector space embeddings of identifiers names, which are at the core of many machine learning models of source code. We compile thousands of ratings gathered from 500 developers into three benchmarks that provide gold standard similarity scores representing the relatedness, similarity, and contextual similarity of identifiers. Using IdBench to experimentally compare five embedding techniques and two string distance functions shows that these techniques differ significantly in their agreement with our gold standard. The best available embedding is very effective at representing how related identifiers are. However, all studied techniques show huge room for improvement in their ability to represent how similar identifiers are. IdBench will help steer future efforts on improved embeddings of identifiers, which will eventually enable better machine learning models of source code.

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
