# OpenReview forum: "Evaluating Semantic Representations of Source Code"
_ICLR.cc/2020/Conference — Reject_

### Official Review · AnonReviewer2 · 2019-10-23
**Official Blind Review #2**

**Rating:** 6

**Review:**

The paper introduces a new dataset that includes manually labelled identifier name pairs. The labels determine how much the corresponding embeddings are useful for determining their meaning in context of code, a setting that has sufficient differences from a natural language.

A significant part of the paper is devoted on data cleaning and relating the computed metrics to similar efforts in natural language processing. While there is not much novelty in this part of the paper, it is doing a good job at addressing many possible questions on the validity of their dataset. Another important aspect that is covered by the work is different kinds of similarities of identifier names - similarity corresponding to having the same or similar type in a precise type system or similarity corresponding to being synonyms. Having several of these dimensions would make the results applicable for a wide range of applications of identifier name embeddings.

While not introducing new concepts, this paper is important for the community, because it has the potential to change the way embedding computation is done for “Big Code” problems. Right now, most papers either introduce their own embeddings, or use non-optimal ones like Code2Vec.

The paper also has a surprising finding that even techniques designed for code are in some cases not as good as the FastText embeddings. This is an interesting result, because few other works include this kind of embedding in their experiments. Furthermore, the paper deep dives into the strong and weak sides of several solutions and shows that there is a large opportunity to improve on the existing results.


**Experience Assessment:**

I have published one or two papers in this area.

**Review Assessment: Checking Correctness Of Derivations And Theory:**

I carefully checked the derivations and theory.

**Review Assessment: Checking Correctness Of Experiments:**

I carefully checked the experiments.

**Review Assessment: Thoroughness In Paper Reading:**

I read the paper thoroughly.

---

> ### Author Response · Authors · 2019-11-09
> **Response to Official Blind Review #2**
>
> Thanks a lot for your insightful review! We are happy to see that the motivation for our work and the contributions of our paper have been made clear.

---

### Official Review · AnonReviewer1 · 2019-10-23
**Official Blind Review #1**

**Rating:** 3

**Review:**

This paper proposed a benchmark dataset for evaluating different embedding methods for identifiers (like variables, functions) in programs. The groudtruth evaluation (similar/related) are labeled via Amazon MTurk. Experiments are carried out on several word embedding methods, and it seems these methods didn’t get good enough correlation with human scores.

Overall I appreciate the effort paid by the authors and human labors. However I have sevarl concerns:

1) why the identifier embedding is important? As pre-trained word embeddings are useful for many NLP downstream tasks, what is the scenario of identifier embedding usage?

2) To collect the human labels, are there any requirements? e.g., experiences in javascript. Especially, I’m curious why in Table 3, setMinutes and setSeconds get score of 0.22 (which is too high).

3) It would make more sense to compare with state of the art language pretraining methods, like bert, xlnet, etc. People have trained the language model with GPT2 (TabNine) that works well with code. So to make the work more convincing, I would suggest to include these.


**Experience Assessment:**

I have read many papers in this area.

**Review Assessment: Checking Correctness Of Derivations And Theory:**

N/A

**Review Assessment: Checking Correctness Of Experiments:**

I assessed the sensibility of the experiments.

**Review Assessment: Thoroughness In Paper Reading:**

I read the paper at least twice and used my best judgement in assessing the paper.

---

> ### Author Response · Authors · 2019-11-09
> **Response to Official Blind Review #1**
>
> Thanks for your review. Please let us address your three concerns:
>
> 1) Importance of identifier embeddings:
> The first four paragraphs of the paper try to answer this question. In short: There are various code-related tasks that recent work has started to address through learning-based techniques, including bug detection, predicting names of methods, predicting types, finding similar code, and automatically fixing bugs. All these techniques rely on a representation of code, which typically is built from representations of identifiers. Simply reusing pre-trained word embeddings from NLP is insufficient, because the vocabulary of source code differs significantly from natural language.
>
> We’d love to receive more specific feedback or suggestions for improving the description of our motivation.
>
> 2) Requirements for humans who contributed the labels:
> Our survey targeted software developers, but did not require knowledge of a specific programming language. To filter uninformed and incorrect ratings, we performed multiple data cleaning steps (Section 2.2). For setMinutes and setSeconds, the reason for a non-zero similarity score presumably is that both functions expect a unit of time as their argument and then store it. The fact that one cannot substitute the other is nicely illustrated by the very low contextual similarity score of 0.06.
>
> 3) Evaluating other models/embeddings, e.g., GPT2 (TabNine):
> Unfortunately, the model (or an embedding derived from it) of TabNine isn’t publicly available (see https://github.com/zxqfl/TabNine). However, if available, then this and any future embeddings can be easily evaluated using or benchmark. If you have concrete pointers to publicly available identifier embeddings, then we'll very happy to include them.

---

### Official Review · AnonReviewer3 · 2019-10-29
**Official Blind Review #3**

**Rating:** 1

**Review:**

This paper presented a crowdsourced dataset for evaluating the semantic relatedness, similarity, and contextual similarity of source code identifiers. Motivated by similar tasks in evaluating the quality of word embeddings in the natural language, authors collected the smalls-scale evaluation datasets (less than three hundreds of code identifier pairs) and further evaluated the performance of several current embeddings techniques for codes or computing the similarity scores of code identifiers. Although I appreciated the efforts to build such evaluation dataset, I think the contributions are limited in terms of scientific contribution.

Pros:

1) They collected a new small-scale evaluation dataset for evaluating the quality of semantic representations of various existing embedding techniques for code identifiers.

2) They performed evaluations on these embeddings techniques for code and provided a few interesting findings based on these tasks.

Cons:

1) The proposed datasets are very small. The total number of code pairs are less than 300 pairs out of total code identifiers 17,000, which is a very small set of the total pairs (17000 x 17000). Therefore, it is hard to fully evaluate the embeddings quality of various methods with high confidence.

2) The whole paper is mainly about the data collection as well as a few of evaluations of several existing code embedding techniques. The scientific contributions are quite limited. It would be nice to put these efforts to have a competition of code embedding techniques and this paper could be served as a technical report on this direction.



**Experience Assessment:**

I have published one or two papers in this area.

**Review Assessment: Checking Correctness Of Derivations And Theory:**

I carefully checked the derivations and theory.

**Review Assessment: Checking Correctness Of Experiments:**

I carefully checked the experiments.

**Review Assessment: Thoroughness In Paper Reading:**

I read the paper thoroughly.

---

> ### Author Response · Authors · 2019-11-09
> **Response to Official Blind Review #3**
>
> Thanks for your review. Here are answers to your two concerns.
>
> 1) Size of dataset:
> The size of the dataset is similar to popular datasets in NLP (Rubenstein & Goodenough RG: 65 pairs, Miller & Charles MC: 30 pairs, Simlex: 999 pairs, WordSim: 353 pairs, MEN: 3000 pairs, but they used a different rating strategy, which made it possible to collect ratings for such a large number of pairs). Since the dataset is gathered from human ratings, obtaining ratings for many more pairs is difficult. Our contribution is not about the size, but about the quality of a benchmark created by human ratings.
>
> Comparison the number of pairs and total identifiers in a corpus is misleading. Large code corpora may have hundreds of thousands of unique identifiers, i.e., using this argument, any number of pairs is “small”. The reason why we sample pairs of identifiers from a large corpus is to cover different domains and different degrees of similarity/relatedness.
>
> 2) Importance of contribution:
> Similar efforts in NLP have served as a catalyst for improved embeddings techniques. Data collection and cleaning is at the heart of creating such benchmarks. As also pointed out by Reviewer 2, having a benchmark is important for the community, and we do not see why an important contribution should be described in a technical report only.

---

### Decision · Program_Chairs · 2019-12-19

**Decision:**

Reject

**Comment:**

This paper presents a dataset to evaluate the quality of embeddings learnt for source code. The dataset consists of three different subtasks: relatedness, similarity, and contextual similarity. The main contribution of the paper is the construction of these datasets which should be useful to the community. However, there are valid concerns raised about the size of the datasets (which is pretty small) and the baselines used to evaluate the embeddings -- there should be a baselines using a contextual embeddings model like BERT which could have been fine-tuned on the source code data. If these comments are addressed, the paper can be a good contribution in an NLP conference. As of now, I recommend a Rejection.